# Deep Imitation Learning for Bimanual Robotic Manipulation

**Fan Xie** †[1]    **Alexander Chowdhury**†[1]    **M. Clara De Paolis Kaluza**[1]

**Linfeng Zhao**[1]    **Lawson L.S. Wong**[1]    **Rose Yu**[1,2] *

## Abstract

We present a deep imitation learning framework for robotic bimanual manipulation in a continuous state-action space. A core challenge is to generalize the manipulation skills to objects in different locations. We hypothesize that modeling the relational information in the environment can significantly improve generalization. To achieve this, we propose to (i) decompose the multi-modal dynamics into elemental movement primitives, (ii) parameterize each primitive using a recurrent graph neural network to capture interactions, and (iii) integrate a high-level planner that composes primitives sequentially and a low-level controller to combine primitive dynamics and inverse kinematics control. Our model is a deep, hierarchical, modular architecture. Compared to baselines, our model generalizes better and achieves higher success rates on several simulated bimanual robotic manipulation tasks. We open source the code for simulation, data, and models at: https://github.com/Rose-STL-Lab/HDR-IL.

## 1   Introduction

Manipulation is a fundamental capability robots need for many real-world applications. Although there has been significant progress in single-arm manipulation tasks such as grasping, pick-and-place, and pushing [1, 2, 3, 4], *bimanual* manipulation has received less attention. Many tasks however require using both arms/hands; consider opening a bottle, steadying a nail while hammering, or moving a table. While having two arms to accomplish these tasks is clearly useful, bimanual manipulation tasks are also significantly more challenging, due to higher-dimensional continuous state-action spaces, more object interactions, and longer-horizon solutions.

Most existing work in bimanual manipulation addresses the problem in a classical control setting, where the environment and its dynamics are known [5]. However, these models are difficult to construct explicitly and are inaccurate, because of complicated interactions in the task including friction, adhesion, and deformation between the two arms/hands and the object being manipulated. A promising approach that avoids manually specifying models is imitation learning, where a teacher provides demonstrations of desired behavior to the robot (sequences of sensed input states and target control actions in those states), and the robot learns a policy to mimic the teacher [6, 7, 8]. In particular, recent deep imitation learning methods have been successful at learning single-arm manipulation from demonstrations, even with only images as observations (e.g., [9, 10]).

In this work, we explore extending deep imitation learning methods to bimanual manipulation. In light of the challenges identified above, our goal is to design an imitation learning model to capture *relational information* (i.e., the trajectory involves relations to other objects in the environment) in environments with *complex dynamics* (i.e., the task requires multiple object interactions to accomplish

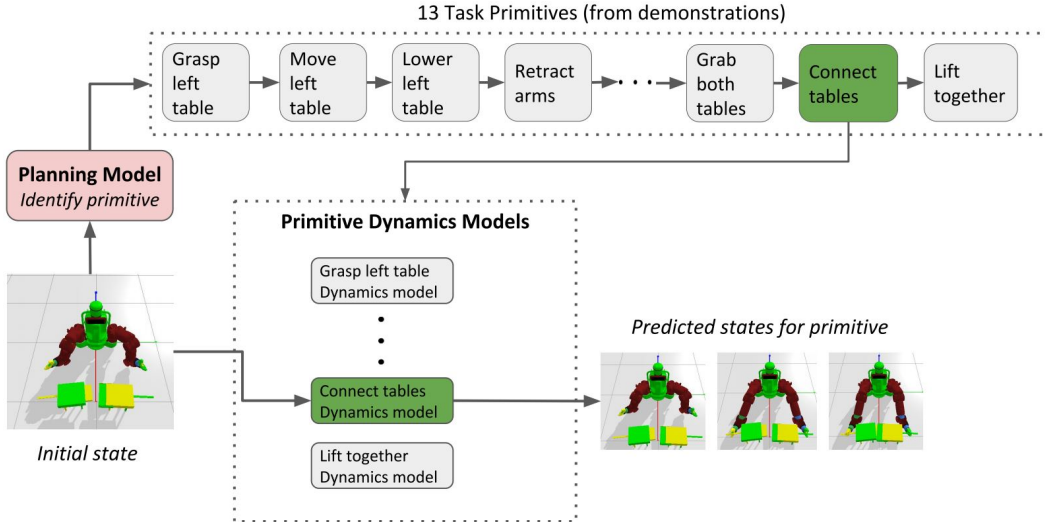

Figure 1: An example of the HDR-ILmodel performing a "peg-in-hole" construction of a table. The trajectory of primitives to complete this task is shown in the top row. The sequence of high-level primitives is learned from demonstrations. State trajectories are predicted based on the primitive identified. The final predicted state for the primitive becomes the initial state for the next primitive.

a goal). The model needs to be general enough to complete tasks under different variations, such as changes in the initial configuration of the robot and objects.

The insight of our paper is to accomplish this goal through a two-level framework, illustrated in Figure 1 for a bimanual "peg-in-hole" table assembly and lifting task. The task involves grasping two halves of a table, slotting one half into the other, and then lifting both parts together as whole. Instead of trying to learn a policy for the entire trajectory, we split demonstrations into task primitives (subsequences of states), as shown in the top row. We learn two models: a high-level *planning model* that predicts a sequence of primitives, and a *set* of low-level *primitive dynamics models* that predict a sequence of robot states to complete each identified task primitive. All models are parameterized by recurrent graph neural networks that explicitly capture robot-robot and robot-object interactions. In summary, our contributions include:

- We propose a deep imitation learning framework, Hierarchical Deep Relational Imitation Learning (HDR-IL), for robotic bimanual manipulation. Our model explicitly captures the relational information in the environment with multiple objects.

- We take a hierarchical approach by separately learning a high-level planning model and a set of low-level primitive dynamics models. We incorporate relational features and use recurrent graph neural networks to parameterize all models.

- We evaluate on two variations of a table-lifting task where bimanual manipulation is essential. We show that both hierarchical and relational modeling provide significant improvement for task competition on held-out instances. For the task shown in Figure 1, our model achieves $29\%$ success rate, whereas a baseline approach without these contributions only succeeds in $1\%$ of the test cases.

## 2   Related Work

The idea of abstracting temporally extended actions has been studied in hierarchical reinforcement learning (HRL) for decades, with the goal of reducing the action search space and sample complexity [11, 12]. A special and useful case is the concept of *parameterized skills*, or *primitives*, where skills are pre-defined and the sequence of skills are learned [13]. Such a hierarchical modeling approach has been widely used in robotics and reinforcement learning [14, 15, 16, 17, 18]. This results in a more structured discrete-continuous action space. The learning for this action space can be naturally decomposed into multiple phases with different primitives applied [19].

Prior work on learning sequences of skills using demonstrations [20, 21, 19, 22, 23] either rely on pre-defined primitive functions or statistical models such as hidden Markov models. Pre-defined primitive functions can be restrictive in representing complex dynamics. Recently, [24, 25] applied pretrained deep neural networks to identify primitives from input states. Hierarchical imitation learning [26, 27, 28] learns the primitives from demonstrations, but does not use relational models.

Our main contribution is learning more accurate dynamics models to support imitation learning of bimanual robotic manipulation tasks. [29] learns a latent representation space to decouple high-level effects and low-level motions, which follows the idea of disentangling the learning into two levels. Recently, graph neural networks (GNNs) [30] have been used to model complex physical dynamics in several applications. Graph models have been used to model physical interaction between objects such as particles [31], [32] and other rigid objects [33], [34]. In the robotics domain, one common approach is to model robots as graphs comprised of nodes corresponding to joints and edges corresponding to links on the robot body, e.g., [35].

## 3   Background

We describe imitation learning using the Markov Decision Process (MDP) framework. An MDP is defined as a tuple $\langle \mathcal{S}, \mathcal{A}, T, R, \gamma \rangle$, where $\mathcal{S}$ is the state space, $\mathcal{A}$ is the action space, $T : \mathcal{S} \times \mathcal{A} \to \Delta(S)$ is the transition function, $R : \mathcal{S} \times \mathcal{A} \to \mathbb{R}$ is the reward function, and $\gamma$ is the discount factor. In imitation learning, we are given a collection of demonstrations $\{\tau^{(i)} = (s_1, a_1, \cdots)\}_{i=1}^{D}$ from an expert policy $\pi_E$, and the objective is to learn a policy $\pi_\theta$ from $\{\tau^{(i)}\}_{i=1}^{D}$ that mimics the expert policy. Imitation learning requires a larger number of demonstrations for long horizon problems, which can be alleviated with hierarchical imitation learning [27].

In hierarchical imitation learning, there is typically a two-level hierarchy: a high-level planning policy and a low-level control policy. The high-level planning policy $\pi_h$ generates a primitive sequence $(p^1, p^2, \cdots)$. In robotic manipulation, a primitive $p^k \in \mathcal{P}$ corresponds to a parameterized policy that maps states to actions $\pi_{p^k} : \mathcal{S} \to \mathcal{A}$, that are typically used to achieve subgoals such as grasping and lifting. Each primitive $p^k$ results in a low-level state trajectory $(s_1^k, s_2^k, \cdots)$. Given a primitive $p^k$ and the initial state $s_t$ , we aim to learn a policy to produce a sequence of states which can be used by an inverse kinematics (IK) solver to obtain the robot control actions $(a_1^k, a_2^k, \cdots)$.

## 4   Methodology

We propose a hierarchical framework for planning and control in robotic bimanual manipulation, outlined in Figure 2. Our framework combines a high-level planning model and a low-level primitive dynamics model to forecast a sequence of $N$ states. We divide the sequence of $N$ states into $K$ primitives $p_1, \cdots, p_K$, with each primitive consisting of a trajectory with a fixed number of states $M = N/K$. A high-level planning model selects the primitive based on previous states $(s_0, ...s_{t-1}) \mapsto p_t$. A low-level primitive dynamics model uses the selected primitive $p_t = p^k$, and predicts the next $M$ steps $s_{t-1} \mapsto (s_t^k ... s_{t+M}^k)$. An inverse kinematics solver then translates a sequence of states into robot control actions $(a_1^k, a_2^k, \cdots)$. We first detail the low-level control model and our contributions to this model. The high-level planning model shares many architectural features with the low-level control model.

### 4.1   Low-level Control Model

Sequence to sequence models [36] with recurrent neural networks (RNN) have been used effectively in prior deep imitation learning methods, e.g. [37]. In our model, we introduce a stochastic component through a variational auto-encoder [38] to generate a latent state distribution $Z_t \sim \mathcal{N}(\mu_{z_t}, \sigma_{z_t})$. The decoder samples a latent state $z_t$ from the distribution and generates a sequence of outputs. We build on this design and innovate in several aspects to execute high-precision bimanual manipulation tasks.

**Relational Features (`Int`)**   The vanilla RNN encoder-decoder model assumes different input features are independent whereas robot arms and objects have strong dependency. To capture these relational dependencies, we introduce a graph attention (GAT) layer [39] in the encoder, leading to graph RNN. We construct a fully connected graph where each node corresponds to each feature

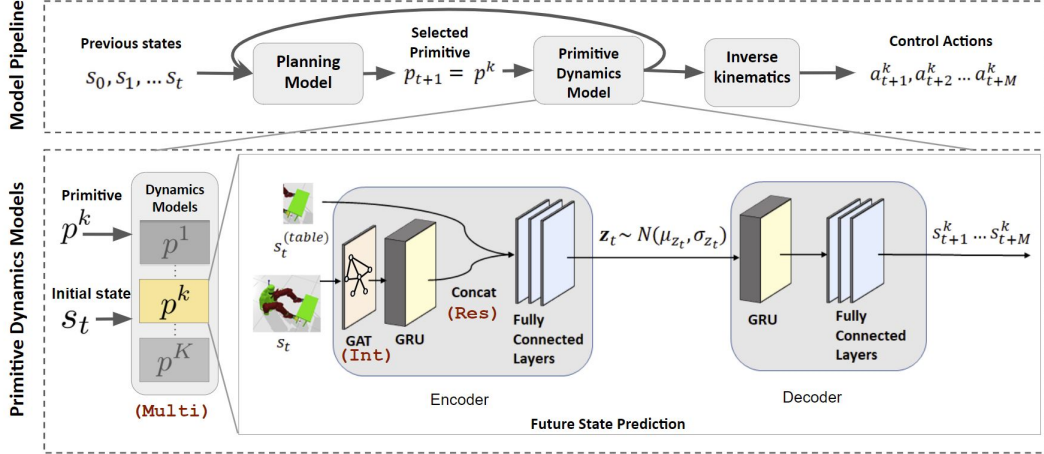

Figure 2: HDR-IL Pipeline: The *planning model* learns sequences of primitives from demonstrations of the task. The *primitive dynamics model* takes an initial state and the learned sequence of primitives to predict future states (robot and object poses) for the task. Finally, we use inverse kinematics to compute low-level robot controls needed to reach the predicted states and execute the commands using the PyBullet physics simulator. The high-level planning allows the identification of primitives, which are used to select the appropriate primitive dynamics model `Multi`. Within each primitive's dynamics model, the GNN layers `Int` capture interactions between objects. The residual connection `Res` of the object state helps direct the projection towards the target. Additional implementation details and hyperparameter settings are found in the supplementary material.

dimension in the state. The attention mechanism of the GAT model learns edge weights through training. Specifically, given two objects features $h_u$ and $h_v$, we compute the edge attention coefficients $e_{uv}$ according to the GAT attention update $e_{uv} = a(\mathbf{W}h_u, \mathbf{W}h_v)$. Here $\mathbf{W}$ are shared trainable weights. These coefficients are used to obtain the attention: $\alpha_{uv} = \text{softmax}(e_{uv})$. The GAT feature update is given by $h_u = \sigma\left(\sum_{v \in \mathcal{N}_u} \alpha_{uv} \mathbf{W} h_v\right)$ where $\mathcal{N}_u$ is the neighbour nodes of $u$. This model allows us to learn parameters that capture the relational information between the objects. The GAT layer that processes the relational features is shown with `Int` in Figure 2.

**Residual Connection (`Res`)**   Another key component of our dynamics model is a residual skip connection that connects features of the target object, such as the table to be moved, to the last hidden layer of the encoder GRU [40]. This idea was inspired by previous robotics work where motor primitives are defined by a dynamics function and a goal state [21]. Skip connections were also used in [41] to help learn complex features in an encoder. The inclusion of the skip connection here helps emphasize the goal, in this case the target object features. This is shown as `Res` in Figure 2.

**Modular Movement Primitives (`Multi`)**   Primitives represent elementary movement patterns such as grasping, lifting and turning. A shared model for all primitive dynamics is limited in its representation power and generalization to more complex tasks. Instead, we take a modular approach at modeling the primitive dynamics. In particular, we design a separate neural network module for each primitive. Each neural network hence captures a particular type of primitive dynamics. We use the same Graph RNN architecture with residual connections for each primitive module. For a task comprised of $K$ ordered primitives with $M$ steps each, each module approximate the primitive policy $\pi_{p^k}$, which produces a state trajectory $(s_1^k, s_2^k, \cdots)$. The last state of each primitive is used as the initial state for the next primitive, which is predicted by the high-level planning model.

**Inverse Kinematics Controller**   In robotics, inverse kinematics (IK) calculates the variable joint parameters needed to place the end of a robot manipulator in a given position and orientation relative to the start. An IK function is used to translate the sequence of states to the control commands for the robot. At each time step, the predicted state for the robot grippers is taken as the input to the IK solver. IK is an iterative algorithm that calculates the robot joint angles needed to reach the desired gripper position. These angles determine the commands on each joint to execute robot movement.

## 4.2 High-level Planning Model

The goal of the high-level planning model is to learn a policy $\pi_h$ which maps a sequence of observed states to a sequence of predefined primitives $\pi_h : \mathcal{S} \rightarrow \mathcal{P}$. Each identified primitive $p^k$ selects the corresponding primitive dynamics policy $\pi_{p^k}$, which is then used for low-level control. Our planning model takes all previous states and infers the next primitive $s_0, s_1, ..s_{t-1} \mapsto p_t$.

Similar to the low-level control model, capturing object interaction in the environment is crucial to accurate predictions of the primitive sequence. Hence, we use the same graph RNN model described in Section 4.1 and introduce relational features `Int` in the input. We omit the residual connections as we did not find them necessary for the primitive identification. An architecture visualization is provided in the Appendix Section A.4.

## 4.3 Supervised Training

The planning model and the control model are trained separately in a supervised learning fashion. We train the high-level *planning model* on manually labeled demonstrations, in order to map a sequence of states $s_1, \cdots, s_{t-1}$ into a primitive label $p_t = k \in [1, \ldots, K]$. The labels are learned with supervision of the correct primitive for the task using the cross entropy loss. The *primitive dynamics model* is trained end-to-end with mean-square error loss on each predicted state $\hat{s}_t$ across all time steps. In the multi-model framework, each primitive is trained with its own parameters for the dynamics model.

# 5 Experiments

We evaluate our approach in simulation using two table lifting tasks which require high precision bimanual manipulation to succeed. We demonstrate the generalization of our model by testing the model with different table starting locations.

## 5.1 Experimental Setup

All simulations for this study were conducted using the PyBullet [42] physics simulator with a Baxter robot. Baxter operates using two arms, each with 7 degrees of freedom. We designed our environment using URDF objects with various weightless links to track location. Our data consists of the $xyz$ coordinates and 4 quaternions for the robot grippers and each object in the environment.

Demonstrations were manually labeled as separate primitives in the simulator and sequenced together to generate the simulation data. Each primitive function was parameterized so they can adapt to different starting and ending coordinates. The primitives were selected such that they each had a distinct trajectory dynamic (ie. lifting upwards vs moving sideways). Each primitive generated a sequence of between 10-12 states for the robot grippers. The number of states were selected through experimentation with the simulation. A minimum number of states were needed for each primitive to produce a smooth non-linear trajectory, while too many states increased the complexity for predictions. We then used the inverse kinematics function to translate the states into command actions for the robot simulator.

**Models** We compare and evaluate several model designs in our experiments. Details about the design elements are in Section 4. The models we consider are:

- GRU Encoder-Decoder Model (`GRU-GRU`): Single model with an GRU encoder, GRU decoder.
- Interaction Network Model (`Int`): This is the `GRU-GRU` model with a fully connected graph network and a graph attention layer, equivalent to feature `Int` in Figure 2.
- Residual Link Model (`Res`): The `GRU-GRU` model with skip connection for the table features, equivalent to feature `Res` in Figure 2.
- Residual Interaction Model (`ResInt`): The `GRU-GRU` model with combined graph structure and skip connection of table object features, equivalent to adding features `Res` and `Int` in Figure 2.
- `HDR-IL` Model: Our multi-model framework which uses multiple `ResInt` models, one model for each primitive. The appropriate model is chosen by the `Multimodule` shown in Figure 2.

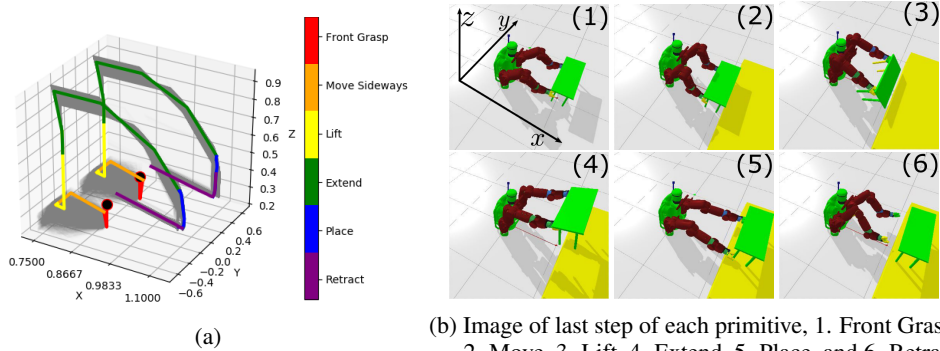

(a)

(b) Image of last step of each primitive, 1. Front Grasp, 2. Move, 3. Lift, 4. Extend, 5. Place, and 6. Retract

Figure 3: Demonstrations for the table lift task. Figure (a) shows the training trajectories in $(x, y, z)$ of left and right grippers for 2500 demonstrations. One sample trajectory is shown in color to highlight the trajectory for each primitive, while the rest are grey. The black dot is the starting location. Figure (b) shows the task executed in our simulator. Each image represents the last step of each primitive. Videos of the demonstrations are provided in the supplementary material.

Table 1: Comparison of model performance by test error and by percent success in 127 test simulations. The Euclidean distance errors are the average of the left and right grippers, and the range represents one standard deviation. Angular distance measures the mean and standard deviation of the geodesic distance between the predicted and actual quaternions. The dynamic time warping distance (DTW) measures similarities between two sequences with a lower value being more similar. The inclusion of the graph and skip connection improves success rates for both the single and multi-model designs.

| Model | Graph | Skip Conn | Multi-Model | Euclidean Dist | Angular Dist | DTW Dist | % Success |
|---|---|---|---|---|---|---|---|
| GRU-GRU | | | | $6.53 \pm 7.05$ | $0.139 \pm 0.182$ | 0.135 | 13% |
| Res | | ✓ | | $7.74 \pm 5.88$ | $0.143 \pm 0.194$ | 0.124 | 13% |
| Int | ✓ | | | $6.67 \pm 5.80$ | $0.145 \pm 0.177$ | 0.123 | 17% |
| ResInt | ✓ | ✓ | | $5.64 \pm 5.17$ | $0.121 \pm 0.205$ | 0.128 | 72% |
| GRU-GRU Multi | | | ✓ | $6.53 \pm 7.05$ | $0.139 \pm 0.182$ | 0.131 | 14% |
| Res Multi | | ✓ | ✓ | $4.97 \pm 5.83$ | $0.123 \pm 0.191$ | 0.121 | 92% |
| Int Multi | ✓ | | ✓ | $11.69 \pm 10.142$ | $0.246 \pm 0.269$ | 0.126 | 13% |
| HDR-IL | ✓ | ✓ | ✓ | $5.01 \pm 5.33$ | $0.112 \pm 0.208$ | **0.119** | **100%** |

## 5.2 Table Lifting Task

**Task Design**   In this task, our robot is tasked to lift a table, with dimensions 35cm by 85cm, onto a platform. We measured success as the table landing upright on the platform. For each demonstration, the table is randomly spawned with the center of the table within a rectangle of 20cm in the $x$ direction and 60cm in the $y$ direction. The task and primitives are illustrated in Figure 3.

**Model Training**   We generate a total of 2,500 training demonstration trajectories, each of length 70. We tested on a random sample of 127 starting points within the range of our training demonstrations. We tune the hyper-parameters using random grid search. All models were trained with the ADAM optimizer with batch size 70 over 12,500 epochs.

**Results**   Table 1 compares the performance of different models. The outputs of model predictions were evaluated using simulation to determine whether predictions were successful in completing the task, with the table ending upright on the platform. For the single-primitive models, we find including *both* the graph and the skip connection together in the ResInt model improves the ability of the model to generalize much more than either feature on its own, as shown in Table 1.

The average Euclidean distance does not necessarily reflect the task performance because percent success largely depends on the model's generalization in the grasp primitive at the beginning of the simulation when the robot attempts to reach the table legs. Analyzing errors by primitive, the large errors are driven by the Extend primitive, which is less crucial for the task. The Extend primitive has the largest step sizes between datapoints, as illustrated in Figure 3. Refer to Table 5 in the Appendix for the Euclidean distances by primitive.

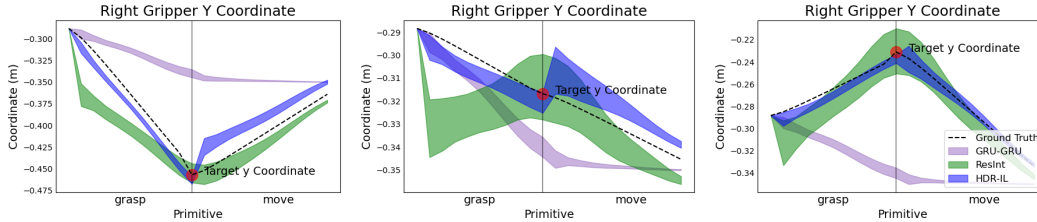

Figure 4: Sample y coordinate predictions for 3 sample demonstrations in the table lifting task. The robot gripper reaches for the table leg, denoted by the "Target", whose location is randomized for each demonstration. The ResInt model has improved ability to generalize, compared to the baseline GRU-GRU model which tends to project an average trajectory over all training demos. The HDR-IL, which uses multiple dynamics models instead of a single model, improves the precision even more.

Table 2: Comparison of model performance between relational data versus absolute data. Absolute data with graphs to measure interactions performs significantly better than relational data when combined with the skip connection.

| Model | Graph | Skip Conn | Relational | Euclidean Dist | Angular Dist | DTW Dist | % Success |
|---|---|---|---|---|---|---|---|
| GRU-GRU | | | | $6.53 \pm 7.05$ | $0.139 \pm 0.182$ | 0.135 | 13% |
| Res Relational | | ✓ | ✓ | $8.15 \pm 4.71$ | $0.171 \pm 0.153$ | 0.133 | 16% |
| ResInt | ✓ | ✓ | | $5.64 \pm 5.17$ | $0.121 \pm 0.205$ | 0.128 | 72% |

Figure 4 visualizes the predictions against the ground truth for the right gripper at the most critical grasp and move primitives. The left gripper follows a similar pattern. We pick the y-coordinates because they have the largest variability in the training demonstrations. The baseline models which did not have a graph structure or skip connections did not generalize well to changes in the table location and project roughly an average of the training trajectories. The ResInt model shows the ability to reach the target, while using multiple models helps to improve upon the precision of the generalizations further.

An alternative method to model object interactions is to use relational coordinates instead of absolute coordinates in the simulation data. We found the use of absolute data and graph structures performs better than using relational data. Combined with the skip connections, the absolute data and graphs performed significantly better in all metrics as shown in Table 2.

We even found our prediction from the HDR-ILmodel can achieve success for table starting locations where the ground truth demonstration failed. There were some failures ($< 0.7\%$ of 2500 demonstrations) due to the approximations of the IK solver sometimes creating unusual movements between arm states. This causes the table to be dropped, which alters the arm trajectories in these demonstrations. Our model can account for these uncertainties in demonstration because of the stochastic sampling design shown in Figure 2.

## 5.3 Peg-in-Hole Task

**Task Design** We evaluate the same models on a more difficult peg-in-hole task. The environment uses two tables, each measuring 35cm by 42.5cm. One table has a peg that needs to fit into the hole of the other table. To lift the tables off the ground, the robot first needs to precisely align and attach the two tables together, as illustrated in Figure 5. The tables start in random locations within their own 20cm by 20cm range. This task introduces three location generalizations, one for the location of each table and one in the location when combining and lifting the joined tables. Videos of the demonstrations are provided in the supplementary material.

**Model Training** The model training follows the same strategy as the table lift task, using a total of 4700 demonstrations, each of length 130, with 13 primitives each of length 10. We trained with a batch size of 130 for 18,800 epochs and tested on a random sample of 281 starting points.

**Results** The results in Table 3 shows our ResInt and HDR-IL model improved success rates compared to the GRU-GRU baseline model. Model success rates were lower across all of the models compared to the table lifting task due to the difficulty of the task. Average Euclidean distance errors

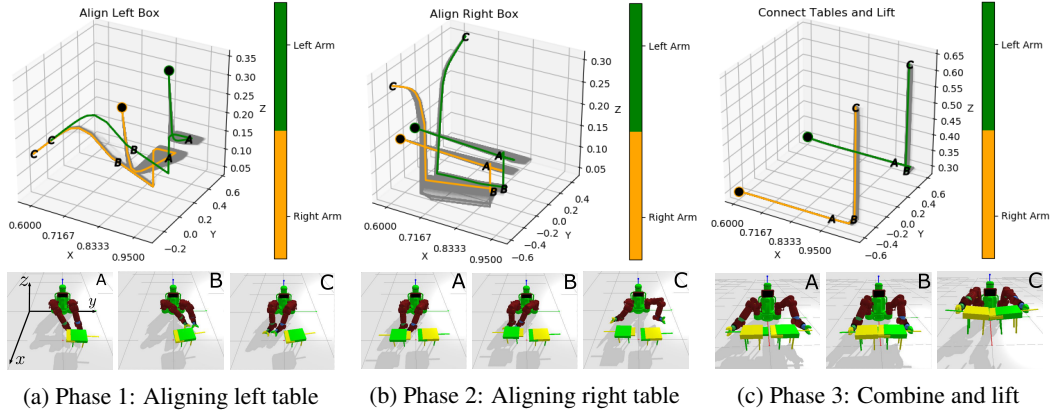

(a) Phase 1: Aligning left table     (b) Phase 2: Aligning right table     (c) Phase 3: Combine and lift

Figure 5: Trajectories of gripper $xyz$ coordinates in the peg-in-hole task. Top row summarizes the actions taking place in 3 phases. The black dot in the 3D figure represents the starting point. The letters correspond to the bottom images. Lift success for this task is defined as both tables elevated evenly above ground, which can only be achieved if the two tables are connected.

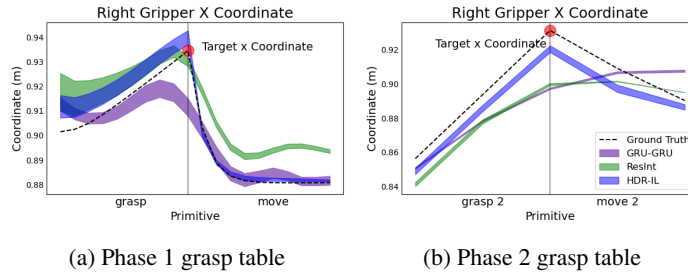

(a) Phase 1 grasp table        (b) Phase 2 grasp table

Figure 6: X coordinate predictions comparison for two phases. (a) the HDR-IL and ResInt models are comparable in their generalization, both reaching the target coordinate, shown as a red circle. (b) 60 time steps into task sequence. HDR-IL shows better generalization compared to the other two models, but it does not reach the target exactly. The decreased accuracy in the second and the third generalizations led to lower success rates in our simulations for all models.

Table 3: Comparison of model performances in the peg-in-hole task. Euclidean distances errors are averages of the two grippers, and the ranges are one standard deviation. Angular distance measures the mean and standard deviation of the geodesic distance between the predicted and actual quaternions. The success rates are lower across all models in the task due to the more difficult task, which is both longer and requires more generalizations. Percent success was measured on 281 test simulations.

| Model | Eculidean Dist | Angular Dist | DTW Dist | % Success - Lift |
|---|---|---|---|---|
| GRU-GRU | $2.11 \pm 1.11$ | $0.029 \pm 0.022$ | 0.121 | 1% |
| ResInt | $1.59 \pm 0.83$ | $0.024 \pm 0.012$ | 0.117 | 15% |
| HDR-IL | $0.90 \pm 1.01$ | $0.013 \pm 0.010$ | **0.113** | **29%** |

are smaller because the range of table starting locations were smaller. The predictions usually fail in phase 2 due to lower accuracy in later time steps.

Generalizations along the x-axis were more difficult for this task due to larger variances in the demonstrations. Figure 6 compares the x-axis coordinate predictions at two different time windows for a single demo. In the time window of the first generalization (a), the HDR-IL and ResInt models are comparable in their generalization. They are both able to reach the target coordinate, shown as a red circle. The second time window (b) corresponding to the generalization in Phase 2 which is the around 60 time steps into task sequence. The HDR-IL model shows better generalization compared to the other two models, but it does not reach the target exactly. The decreased accuracy in the second and the third generalizations led to lower success rates in our simulations for all models.

# 6    Conclusion and Future Work

We present a novel deep imitation learning framework for learning complex control policies in bimanual manipulation tasks. This framework introduces a two-level hierarchical model for primitive selection and primitive dynamics modeling, utilizing graph structures and residual connections to model object dynamics interactions. In simulation, our model shows promising results on two complex bimanual manipulation tasks: table lifting and peg-in-hole. Future work include estimating object pose directly from visual inputs. Another interesting direction is automatic primitive discovery, which would greatly improve the label efficiency of training the model in new tasks.

## Broader Impact

Robotics systems that utilize fully automated policies for different tasks have already been applied to many manufacturing, assembly lines, and warehouses processes. Our work demonstrates the potential to take this automation one step further. Our algorithm can automatically learn complex control policies from expert demonstrations, which could potentially allow robots to augment their existing control designs and further optimize their workflows. Implementing learned policies in safety-critical environments such as large-scale assembly lines can be risky as these algorithms do not have guaranteed precision. Improved theoretical understanding and interpretability of model policies could potentially mitigate these risks.

## Acknowledgments and Disclosure of Funding

This work was supported in part by the U. S. Army Research Office under Grant W911NF-20-1-0334. Fan Xie, Alex Chowdhury, and Linfeng Zhao were supported by the Khoury Graduate Research Fellowship and the Khoury seed grant at Northeastern University. Additional revenues related to this work: NSF #1850349, ONR # N68335-19-C-0310, Google Faculty Research Award, Adobe Data Science Research Awards, GPUs donated by NVIDIA, and computing allocation awarded by DOE.

## Footnotes

*†Equal contribution; Correspondence: xie.f@northeastern.edu,chowdhury.al@northeastern.edu, roseyu@ucsd.edu; [1]Northeastern University, Boston MA, USA, [2]University of California, San Diego, USA.

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
