[Supplementary Material 1]

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

# A Model Details

## A.1 Model Parameters

The Hyper-parameters we tested and the best parameter we selected for each model.

Table 4: Summary of model hyperparameters tested in hyperparameter search and the value used in the final model.

| Table Lift Task | | | | |
|---|---|---|---|---|
| Model | Parameters Searched | Planning Model | Dynamics Model - Single | Dynamics Model - Multi |
| Encoder LR | $2 \times 10^{-4}$ - $1 \times 10^{-5}$ | $5 \times 10^{-5}$ | $2 \times 10^{-4}$ | $1 \times 10^{-5}$ |
| Decoder LR | $2 \times 10^{-4}$ - $1 \times 10^{-5}$ | $5 \times 10^{-5}$ | $4 \times 10^{-5}$ | $4 \times 10^{-5}$ |
| Hidden Dimensions | 100 - 1024 | 512 | 512 | 512 |
| Encoder FC Layers | 3-18 | 3 | 18 | 3 |
| Decoder FC Layers | 3-19 | 4 | 19 | 5 |
| GAT attention layers | 1, 2 | 1 | 1 | 1 |
| Peg-In-Hole Task | | | | |
| Model | Parameters Searched | Planning Model | Dynamic Model - Single | Dynamic Model - Multi |
| Encoder LR | $2 \times 10^{-4}$ - $1 \times 10^{-5}$ | $1 \times 10^{-5}$ | $5 \times 10^{-5}$ | $5 \times 10^{-5}$ |
| Decoder LR | $2 \times 10^{-4}$ - $8 \times 10^{-7}$ | $8 \times 10^{-7}$ | $5 \times 10^{-5}$ | $5 \times 10^{-5}$ |
| Hidden Dimensions | 100 - 1024 | 512 | 1024 | 512 |
| Encoder FC Layers | 3-20 | 9 | 20 | 3 |
| Decoder FC Layers | 3-21 | 8 | 21 | 5 |
| GAT attention layers | 1, 2 | 1 | 1 | 1 |
| GAT attention heads | 1, 2 | 1 | 1 | 1 |

## A.2 Data Pre-processing

For the table lift task, we used 2,500 training demonstrations, with starting locations distributed uniformly over displacement range of 20cm by 60 cm. The testing was done on 127 random starting locations with displacements within the range of the training dataset. For the peg-in-hole task we used 4,700 randomly selected demonstrations for training and 281 random locations for testing. In data pre-processing, we removed demonstrations that were not successful in completing the tasks.

## A.3 Dynamic Model Details

### A.3.1 Encoder

Figure 7: Encoder training set-up details with graph structure and skip connection.

The baseline encoder consists of a GRU which takes the initial state as the input. The number of layers in the GRU corresponds to the number of states being projected. The initial state is a vector consisting of the $x$-, $y$-, and $z$-coordinates of each object concatenated with the quaternions for that object. During training, we use teacher forcing so that the input in each layer of the GRU take the states from the simulation. During testing, the inputs in layers besides the first come from the output

of the previous layer. The hidden state of the GRU at the last time point is passed through the fully connected layers. The number of fully connected layers vary based on whether the model is part of a multi-model framework as outlined in the Model Parameters section. We modified the number of linear layers such that our single and multi-model designs have comparable number of parameters. The final layer doubles the dimensionality of the fully connected layers, splitting the hidden state into mean and variances for the decoder.

**Relational Model** `Int`   Models that use relational features use graph attention layers (GAT). We use one attention head in the GAT layers.

A node corresponds to an object feature in the state, one node for each of the x, y, z coordinates and for the four quaternions. We will index the layer with a superscript. The initial node feature $h_u^{(0)}$ is the value of that feature. For each layer, the edge attention coefficient $e_{uv}^l$ between nodes $u$ and $v$ is given by

$$e_{uv}^{(l)} = a\left(\mathbf{W}^{(l)}h_u^{(l)}, \mathbf{W}^{(l)}h_v^{(l)}\right) \tag{1}$$

for learned layer weight matrix $\mathbf{W}^{(l)}$. The attention mechanism $a$ concatenates the inputs and multiplies them by a learned weight vector $\mathbf{a}^{(l)}$ and applies a nonlinearity, i.e.

$$e_{uv}^{(l)} = \text{LeakyReLU}\left(\mathbf{a}^{(l)}(\mathbf{W}^{(l)}h_u^{(l)}||\mathbf{W}^{(l)}h_v^{(l)})\right) \tag{2}$$

Attention weights are computed by normalizing edge attention coefficients with the softmax function

$$\alpha_{uv} = \text{softmax}(e_{uv}). \tag{3}$$

Node features are aggregated by neighborhood. For node $u$ and the set of its connected neighbors $\mathcal{N}_u$, node features are updated by the GAT update rule,

$$h_u^{(l+1)} = \sum_{v \in \mathcal{N}_u} \alpha \mathbf{W}^{(l)} h_v^{(l)}. \tag{4}$$

Figure 8: A graphical visualization of the attention weights learned in the 2nd layer of the GRU for the table lifting task. Node colors represent different objects in the environment. In this graph, 'R' and 'L' stand for left and right grippers, and 'T' for table. The visible edges are those with attention weights greater than 8%. A graph convolution network (GCN) without learned weights would assume uniform attention weights of 4.77% for all 21 nodes. This supported our decision to use GAT over GCN for modeling the interactions. The two nodes receiving the largest weights are the right gripper y coordinate and a quaternion on the left gripper.

**Residual Connection** `Res`   Models that have the residual connection concatenate the table features to the final hidden state from the GRU, prior to the linear layers.

### A.3.2   Decoder

Figure 9: Decoder training set-up details.

The decoder takes a sample latent space from the encoder. The sample is the initial hidden state going into the GRU of the decoder. The initial input of the first layer is a tensor of zeros. In the remaining layers, the input to the layer is the output from the previous layer. At each layer, the output of the GRU goes through a series of fully connected layers to get the final output of the model which represent the states at each time point.

**ODE Model**   In the ODE model, the latent state is passed to the ODE Solver, which replaces the GRU. The ODE solver solves for the time invariant gradient function, which used to generate the output at each time step.

### A.4   Planning Model Details

Figure 10: Planning Models take a sequence of observed states and determines a primitive for the next step. The encoder setup is the same as the dynamics model, with the graph structure but no skip connection. The latent state in fixed rather than a distribution in the dynamic model. In the decoder, the output of the GRU and fully connected layers are fed through an additional softmax layer to get the primitive label.

## B  Additional Results

Figure 11: **RNN vs ODE Model** Introducing continuous time dynamics through the ODE model did not improve upon the predictions. This is due to the transitions of different primitives in this task being discrete.

Table 5: Breakdown of average Euclidean distance (cm) by primitives of table lift task. Errors are one standard deviation. The errors are the largest in the lift primitive because this primitive has the largest distances between time steps. The larger movements between time steps results in less precision in the predictions

| Model | Grasp | Move | Lift | Extend | Place | Retract |
|---|---|---|---|---|---|---|
| GRU-GRU | $7.03 \pm 5.45$ | $9.34 \pm 6.59$ | $2.67 \pm 1.66$ | $14.20 \pm 9.94$ | $4.65 \pm 2.91$ | $1.39 \pm 0.39$ |
| Res | $8.33 \pm 5.64$ | $9.62 \pm 6.69$ | $4.44 \pm 1.35$ | $13.90 \pm 7.74$ | $5.30 \pm 0.75$ | $4.94 \pm 1.41$ |
| Int | $7.73 \pm 5.20$ | $9.66 \pm 6.37$ | $3.44 \pm 1.19$ | $12.57 \pm 6.91$ | $4.23 \pm 2.40$ | $2.61 \pm 0.71$ |
| ResInt | $3.80 \pm 1.89$ | $3.28 \pm 1.36$ | $3.87 \pm 1.55$ | $13.46 \pm 8.09$ | $5.72 \pm 2.34$ | $3.45 \pm 0.69$ |
| GRU-GRU Multi | $7.03 \pm 5.45$ | $9.35 \pm 6.59$ | $2.68 \pm 1.65$ | $14.20 \pm 9.94$ | $4.65 \pm 2.92$ | $1.40 \pm 0.38$ |
| Res Multi | $3.76 \pm 1.63$ | $3.59 \pm 1.49$ | $3.09 \pm 1.87$ | $13.53 \pm 9.62$ | $4.08 \pm 2.22$ | $1.54 \pm 0.61$ |
| Int Multi | $10.99 \pm 9.20$ | $10.01 \pm 8.05$ | $3.35 \pm 2.34$ | $18.04 \pm 10.42$ | $6.94 \pm 4.24$ | $20.68 \pm 10.78$ |
| HDR-IL | $2.86 \pm 1.51$ | $3.29 \pm 1.34$ | $2.98 \pm 1.87$ | $12.78 \pm 8.71$ | $3.73 \pm 2.85$ | $4.03 \pm 0.85$ |

Table 6: Breakdown of average Euclidean distance (cm) by phase for the peg-in-hole task. Errors are one standard deviation.

| Model | Phase 1 | Phase 2 | Phase 3 |
|---|---|---|---|
| GRU-GRU | $1.89 \pm 1.08$ | $2.51 \pm 1.22$ | $1.79 \pm 0.67$ |
| ResInt | $1.93 \pm 0.85$ | $1.49 \pm 0.86$ | $1.18 \pm 0.42$ |
| HDR-IL | $1.45 \pm 1.24$ | $1.79 \pm 1.30$ | $1.29 \pm 0.66$ |

## C  Simulations

All tasks used to train and test our framework were designed using the PyBullet physics simulator. Our goal when designing tasks was two-fold:

- Design tasks that could be easily decomposed into a sequence of simpler, low-level primitives.
- Design tasks that could not be performed using only one arm, and thus would fall under the domain of bimanual manipulation.

**Task Design**  Designing both tasks used for our experiments followed the same process. We first decided which low-level primitives the task should be decomposed into. Then we manually coded the primitives required and put them together sequentially. We made sure that if the robot used only one arm for either of the tasks it would fail, to ensure that learning to do the task successfully required true bimanual manipulation. We used the final position of the center of mass of the table or tables to measure task success.

**Primitive Design**   The process of designing the primitives was iterative. We first built the primitive movements and test them on the simulation of the task with the table at various starting locations and observe the performance and success rates. We observe where the simulations failed and adjust the primitives parameters as necessary. This including modifying the number of time steps and the trajectory of the primitive to avoid accidental interactions. All code used for our simulations will be made publicly available.

**Datasets**   All data was collected using the two scripts designed for the place-and-lift and peg-in-hole tasks. Both tasks had noise introduced to them in order to make the training data more robust. Our code is included to run our simulations and generate datasets.

[Supplementary Material 2]

# A  Model Details

## A.1  Model Parameters

The Hyper-parameters we tested and the best parameter we selected for each model.

Table 1: Summary of model hyperparameters tested in hyperparameter search and the value used in the final model.

| Model | Parameters Searched | Planning Model | Dynamics Model - Single | Dynamics Model - Multi |
|---|---|---|---|---|
| **Table Lift Task** | | | | |
| Encoder LR | $2 \times 10^{-4}$ - $1 \times 10^{-5}$ | $5 \times 10^{-5}$ | $2 \times 10^{-4}$ | $1 \times 10^{-5}$ |
| Decoder LR | $2 \times 10^{-4}$ - $1 \times 10^{-5}$ | $5 \times 10^{-5}$ | $4 \times 10^{-5}$ | $4 \times 10^{-5}$ |
| Hidden Dimensions | 100 - 1024 | 512 | 512 | 512 |
| Encoder FC Layers | 3-18 | 3 | 18 | 3 |
| Decoder FC Layers | 3-19 | 4 | 19 | 5 |
| GAT attention layers | 1, 2 | 1 | 1 | 1 |
| **Peg-In-Hole Task** | | | | |
| Model | Parameters Searched | Planning Model | Dynamic Model - Single | Dynamic Model - Multi |
| Encoder LR | $2 \times 10^{-4}$ - $1 \times 10^{-5}$ | $1 \times 10^{-5}$ | $5 \times 10^{-5}$ | $5 \times 10^{-5}$ |
| Decoder LR | $2 \times 10^{-4}$ - $8 \times 10^{-7}$ | $8 \times 10^{-7}$ | $5 \times 10^{-5}$ | $5 \times 10^{-5}$ |
| Hidden Dimensions | 100 - 1024 | 512 | 1024 | 512 |
| Encoder FC Layers | 3-20 | 9 | 20 | 3 |
| Decoder FC Layers | 3-21 | 8 | 21 | 5 |
| GAT attention layers | 1, 2 | 1 | 1 | 1 |
| GAT attention heads | 1, 2 | 1 | 1 | 1 |

## A.2  Data Pre-processing

For the table lift task, we used 2,500 training demonstrations, with starting locations distributed uniformly over displacement range of 20cm by 60 cm. The testing was done on 127 random starting locations with displacements within the range of the training dataset. For the peg-in-hole task we used 4,700 randomly selected demonstrations for training and 281 random locations for testing. In data pre-processing, we removed demonstrations that were not successful in completing the tasks.

## A.3  Dynamic Model Details

### A.3.1  Encoder

Figure 1: Encoder training set-up details with graph structure and skip connection.

The baseline encoder consists of a GRU which takes the initial state as the input. The number of layers in the GRU corresponds to the number of states being projected. The initial state is a vector consisting of the $x$-, $y$-, and $z$-coordinates of each object concatenated with the quaternions for that object. During training, we use teacher forcing so that the input in each layer of the GRU take the states from the simulation. During testing, the inputs in layers besides the first come from the output

of the previous layer. The hidden state of the GRU at the last time point is passed through the fully connected layers. The number of fully connected layers vary based on whether the model is part of a multi-model framework as outlined in the Model Parameters section. We modified the number of linear layers such that our single and multi-model designs have comparable number of parameters. The final layer doubles the dimensionality of the fully connected layers, splitting the hidden state into mean and variances for the decoder.

**Relational Model `Int`** Models that use relational features use graph attention layers (GAT). We use one attention head in the GAT layers.

A node corresponds to an object feature in the state, one node for each of the x, y, z coordinates and for the four quaternions. We will index the layer with a superscript. The initial node feature $h_u^{(0)}$ is the value of that feature. For each layer, the edge attention coefficient $e_{uv}^l$ between nodes $u$ and $v$ is given by

$$e_{uv}^{(l)} = a\left(\mathbf{W}^{(l)} h_u^{(l)}, \mathbf{W}^{(l)} h_v^{(l)}\right) \tag{1}$$

for learned layer weight matrix $\mathbf{W}^{(l)}$. The attention mechanism $a$ concatenates the inputs and multiplies them by a learned weight vector $\mathbf{a}^{(l)}$ and applies a nonlinearity, i.e.

$$e_{uv}^{(l)} = \text{LeakyReLU}\left(\mathbf{a}^{(l)}(\mathbf{W}^{(l)} h_u^{(l)} \| \mathbf{W}^{(l)} h_v^{(l)})\right) \tag{2}$$

Attention weights are computed by normalizing edge attention coefficients with the softmax function

$$\alpha_{uv} = \text{softmax}(e_{uv}). \tag{3}$$

Node features are aggregated by neighborhood. For node $u$ and the set of its connected neighbors $\mathcal{N}_u$, node features are updated by the GAT update rule,

$$h_u^{(l+1)} = \sum_{v \in \mathcal{N}_u} \alpha \mathbf{W}^{(l)} h_v^{(l)}. \tag{4}$$

Figure 2: A graphical visualization of the attention weights learned in the 2nd layer of the GRU for the table lifting task. Node colors represent different objects in the environment. In this graph, 'R' and 'L' stand for left and right grippers, and 'T' for table. The visible edges are those with attention weights greater than 8%. A graph convolution network (GCN) without learned weights would assume uniform attention weights of 4.77% for all 21 nodes. This supported our decision to use GAT over GCN for modeling the interactions. The two nodes receiving the largest weights are the right gripper y coordinate and a quaternion on the left gripper.

34 **Residual Connection** `Res`    Models that have the residual connection concatenate the table features
35 to the final hidden state from the GRU, prior to the linear layers.

36 **A.3.2    Decoder**

Figure 3: Decoder training set-up details.

37 The decoder takes a sample latent space from the encoder. The sample is the initial hidden state
38 going into the GRU of the decoder. The initial input of the first layer is a tensor of zeros. In the
39 remaining layers, the input to the layer is the output from the previous layer. At each layer, the output
40 of the GRU goes through a series of fully connected layers to get the final output of the model which
41 represent the states at each time point.

42 **ODE Model**    In the ODE model, the latent state is passed to the ODE Solver, which replaces the
43 GRU. The ODE solver solves for the time invariant gradient function, which used to generate the
44 output at each time step.

45 **A.4    Planning Model Details**

Figure 4: Planning Models take a sequence of observed states and determines a primitive for the
next step. The encoder setup is the same as the dynamics model, with the graph structure but no skip
connection. The latent state in fixed rather than a distribution in the dynamic model. In the decoder,
the output of the GRU and fully connected layers are fed through an additional softmax layer to get
the primitive label.

## B    Additional Results

Figure 5: **RNN vs ODE Model** Introducing continuous time dynamics through the ODE model did not improve upon the predictions. This is due to the transitions of different primitives in this task being discrete.

Table 2: Breakdown of average Euclidean distance (cm) by primitives of table lift task. Errors are one standard deviation.

| Model | Grasp | Move | Lift | Extend | Place | Retract |
|---|---|---|---|---|---|---|
| GRU-GRU | $7.03 \pm 5.45$ | $9.34 \pm 6.59$ | $2.67 \pm 1.66$ | $14.20 \pm 9.94$ | $4.65 \pm 2.91$ | $1.39 \pm 0.39$ |
| Res | $8.33 \pm 5.64$ | $9.62 \pm 6.69$ | $4.44 \pm 1.35$ | $13.90 \pm 7.74$ | $5.30 \pm 0.75$ | $4.94 \pm 1.41$ |
| Int | $7.73 \pm 5.20$ | $9.66 \pm 6.37$ | $3.44 \pm 1.19$ | $12.57 \pm 6.91$ | $4.23 \pm 2.40$ | $2.61 \pm 0.71$ |
| ResInt | $3.80 \pm 1.89$ | $3.28 \pm 1.36$ | $3.87 \pm 1.55$ | $13.46 \pm 8.09$ | $5.72 \pm 2.34$ | $3.45 \pm 0.69$ |
| GRU-GRU Multi | $7.03 \pm 5.45$ | $9.35 \pm 6.59$ | $2.68 \pm 1.65$ | $14.20 \pm 9.94$ | $4.65 \pm 2.92$ | $1.40 \pm 0.38$ |
| Res Multi | $3.76 \pm 1.63$ | $3.59 \pm 1.49$ | $3.09 \pm 1.87$ | $13.53 \pm 9.62$ | $4.08 \pm 2.22$ | $1.54 \pm 0.61$ |
| Int Multi | $10.99 \pm 9.20$ | $10.01 \pm 8.05$ | $3.35 \pm 2.34$ | $18.04 \pm 10.42$ | $6.94 \pm 4.24$ | $20.68 \pm 10.78$ |
| HDR-IL | $2.86 \pm 1.51$ | $3.29 \pm 1.34$ | $2.98 \pm 1.87$ | $12.78 \pm 8.71$ | $3.73 \pm 2.85$ | $4.03 \pm 0.85$ |

Table 3: Breakdown of average Euclidean distance (cm) by phase for the peg-in-hole task. Errors are one standard deviation.

| Model | Phase 1 | Phase 2 | Phase 3 |
|---|---|---|---|
| GRU-GRU | $1.89 \pm 1.08$ | $2.51 \pm 1.22$ | $1.79 \pm 0.67$ |
| ResInt | $1.93 \pm 0.85$ | $1.49 \pm 0.86$ | $1.18 \pm 0.42$ |
| HDR-IL | $1.45 \pm 1.24$ | $1.79 \pm 1.30$ | $1.29 \pm 0.66$ |

## C    Simulations

All tasks used to train and test our framework were designed using the PyBullet physics simulator. Our goal when designing tasks was two-fold:

- Design tasks that could be easily decomposed into a sequence of simpler, low-level primitives.
- Design tasks that could not be performed using only one arm, and thus would fall under the domain of bimanual manipulation.

**Task Design**    Designing both tasks used for our experiments followed the same process. We first decided which low-level primitives the task should be decomposed into. Then we manually coded the primitives required and put them together sequentially. We made sure that if the robot used only one arm for either of the tasks it would fail, to ensure that learning to do the task successfully required true bimanual manipulation. We used the final position of the center of mass of the table or tables to measure task success.

**Primitive Design**    The process of designing the primitives was iterative. We first built the primitive movements and test them on the simulation of the task with the table at various starting locations

and observe the performance and success rates. We observe where the simulations failed and adjust the primitives parameters as necessary. This including modifying the number of time steps and the trajectory of the primitive to avoid accidental interactions. All code used for our simulations will be made publicly available.

**Datasets** All data was collected using the two scripts designed for the place-and-lift and peg-in-hole tasks. Both tasks had noise introduced to them in order to make the training data more robust. Our code is included to run our simulations and generate datasets.

[Supplementary Material 3]

**Description of Files:**
- Projection Models - Contains the models used for projection. The models are labeled based on their description in the paper.
- Results - contains the testing demonstration results for each model.
- Simulation Code - contains the model used to run simulations and generate data for our experiment
- Simulations.mp4 – This video shows our task demonstration compared with the learned model prediction. We also show some attempts at lifting the table with one arm to demonstrate that the table requires both arms to lift, and failure cases where the peg-in-hole are not inserted properly before lifting.

**Model Dependencies:**

Simulation Models
- bumpy
- pandas
- pybullet

Projection Models
- statistics
- torch
- pandas
- numpy
- random
- csv
- dgl

**Projection Model Runtime:**

Models were trained on a laptop with an i7-8750H processor and Nvidia 1050Ti Max-Q GPU.
2500 demonstrations for the table lift task takes about 2 hours to run
4500 demonstrations for the peg-in-hole task takes about 3.5 hours to run

**Projection Model Instructions:**

Training code and evaluation code are saved in the 'Model' folder. Training and evaluation are contained within the same model.
Within each model, there are 7 files relevant to the model.

1. Main - runs all of the models to get the projection
2. A1PrimitiveData - Data reading for planning
3. A2SoftmaxModel - Prediction models for planning
4. A3TrainSoftmax - Data processing and training/evaluation for planning
5. B1DataProcessing - Data reading for dynamic model
6. B2ODEModel - Prediction models for dynamic model

7. B3TrainODE - Data processing and training/evaluation for planning

To run the models, run the Main file in each model. There are a few modifications that are made to get either the training or testing data.

1. In the A1 and B1 files, make sure the directory in the BaxterDataset constructor is pointing to the correct CSV file.

2. A3 - The Options can be modified for the run. Description of the options are in the comments. The options here decide whether to load the previous model or train a new model.

3. B3 - The Options can be modified for the run. Description of the options are in the comments. The options here decide whether to load the previous model or train a new model.

4. Main - Options can be modified for the run. Run this to generate the projections.