[Reviews · NeurIPS 2020]

Review 1

Summary and Contributions: This paper presents a two stage hierarchical imitation learning framework where the high-level planner predicts the next primitive to be executed from the previous history of state sequence and the low-level planner predicts the next sequence of states for a fixed horizon. A task is represented by a sequence of movement primitives and each movement primitive is encoded with a graph recurrent neural network designed to capture robot-object relational features in the environment. The proposed methodology is evaluated on table lifting and peg-in-hole tasks suggesting performance improvement in capturing relational dependencies with graph RNN, residual skip connections and use of multiple primitives.

Strengths: Bimanual coordination for robotic manipulation is an important venue of research. Utilizing graph structures in robot manipulation is of growing interest to a wider robotics community. The main contribution of the paper is to learn accurate dynamic models that can be useful for bimanual robot manipulation. The accuracy is achieved using Graph RNN, feeding the goal-directed object as residual connection and using multiple primitives that are sequenced together to execute the task.

Weaknesses: My main concerns are: - Feeding the actual pose of one arm (master) and the relative pose of the second arm (slave) with respect to the master and similarly other objects would have been more informative for the network to capture the relational dependencies at the pose level. A baseline comparison with this method would be useful to understand the dependency structure, especially to improve the performance for the second task. Adding other baselines with state-of-the-art methods in the related work would further improve the understanding of the work. - The authors discuss a few examples with the position in table tasks, but the effect of orientation is not explained. Does the approach generalize with randomly sampled orientation of the target object ? Are the orientations normalized to unit quaternions after prediction ? The authors are encouraged to show orientation errors and quantify the performance. - Adding visual pixel information from pixels would help to establish the true merits of graph attention mechanism. - 29 percent accuracy with table assembly tasks is rather low. The Euclidean distance error units in Table 1 seem very high. Are they normalized to per datapoint position errors ? If so, an error of 5 cm with HDR-IL seems unreasonably high. - It is not clear if the proposed methodology is specific to bimanual manipulation. Just using robotic manipulation could be more appropriate. - Experiments with real setup would have been useful to establish the merits of the proposed approach.

Correctness: It seems to me the main contribution is use of graph RNN, residual connection of target object and multiple primitives; along with preliminary experiments for bi-manual manipulation. The contributions claimed in the introduction are rather generic which are common to several other papers including the ones mentioned in the related work. It is not entirely clear to me how effectively the graph attention mechanism captures the relational dynamics between the objects and the scene. A simpler experiment which encodes the relational structure between objects and/or the end-effectors, in particular the effect of orientation, would have been useful to establish the claim of the paper.

Clarity: The paper is well-written and easy to follow for the most part. The experimental set-up is sound and the performance metrics are intuitive to understand.

Relation to Prior Work: The related work discusses hierarchical reinforcement learning and sequencing primitives. It is also related to activity recognition in computer vision. Other useful references: [1] Learning bimanual end-effector poses from demonstrations using task-parameterized dynamical systems, IROS 2015 [2] Motion2Vec: Semi-supervised representation learning from surgical videos, ICRA 2020

Reproducibility: Yes

Additional Feedback: There is a good potential scope of this work, and the preliminary results are encouraging. The authors are encouraged to further ground their claims by comparing their approach with other state-of-the-art methods in the literature. Thanks for clarifying about adding relative poses baseline in the rebuttal. I would encourage the authors to strengthen the evaluation experiments and the baselines for the final version.


Review 2

Summary and Contributions: This work presents an imitation learning method, HDR-IL, for bimanual robotic manipulation. They propose a hierarchical modular network with a high level planner that predicts a sequence of primitive tasks and a low level dynamics model. Their main contributions are using the hierarchical and relational imitation learning model to solve bimanual manipulation on 2 tasks: table lifting and peg-in-hole. They claim to have better generalization and higher success rates when using hierarchy and relational features.

Strengths: The high-level planner / low level controller design and pipeline is explained pretty clearly in the methodology section and each section of the model is explained in detail. The paper is hence very well written. In general, hierarchical control offers several benefits, which has been well discussed and motivated in this paper. This line of work and its applications to bimanual manipulation hasn’t been explored much and this work shows that their method succeeds on 2 tasks that require two arms. I also appreciate that the different failure cases were included in the videos in addition to cases where it worked. Furthermore, the method is well ablated, showing that all components are needed for the proposed method to work.

Weaknesses: In terms of background and related work, it would be nice to see more explanations of current cited methods. What kinds of methods do current bimanual robotic manipulation tasks use? Only saw one sentence that they’re done in the classical control setting. The baseline in the experiments has a 1% success rate, are there success rates that can be compared to in different methods for bimanual manipulation? No actual baselines. The claimed baselines are pretty much ablations of the proposed method. Implementations of previous cited work would be needed to judge the efficacy of specifying the primitives the way it is. For the table task: “success for table starting locations where the ground truth demonstration failed.” Some explanation on when/why did the ground truth demonstration fail, when the HDR-IL method succeeds, and how do you get ground truths? In the conclusion, it’s stated that “our pipeline begins by manually designing and labeling primitives to train the high-level model”, curious to see how these primitives are labeled. “Incorporate an explicit graph structure to model interactions” (54). So the method seems very limited to the provision of the graph structure. It would be good to see a discussion on how to automatically learn the graph structure. The experiments and method do not show off how complex two handed settings can be: “These models are difficult to construct explicitly … complicated interactions in the task including friction, adhesion, and deformation between the two arms/hands and the object being manipulated” (25). But the experiments are all in simulation, with “various weightless links to track location” (180), nor is it mentioned that the table is flexible. Better experiments would be holding a nail to hammer down (as mentioned in line 20). It would be more interesting if the table was to land in a specified configuration, such as on its side / flips, which would require (presumably) more arm to arm interaction / model learning. Peg insertion task -- it is unclear why the table is needed to be lifted if this is the case, it seems reasonable that the task can be done with 1 arm, and 2 is not necessary. Unclear how the primitives are constructed. Seems like the states used to map are manually crafted / designed (184, 60 in appendix), which is contradicting to “ learns a high level … and a set of low-level primitive dynamics models” (53)

Correctness: Overall, looks correct.

Clarity: Overall, easy to read. More comments on clarity are in the strength and weaknesses section.

Relation to Prior Work: Overall, good. More comments on clarity are in the strength and weaknesses section.

Reproducibility: No

Additional Feedback: Clarity on how the primitives are manually annotated is missing. This is needed for reproduction. I have read all the other reviews and the author feedback. Thank you for addressing my concerns. I'm hence increasing my score. However, I still have reservations on the experimental setup and would encourage the authors to strengthen this paper with thorough experimentation and comparisons.


Review 3

Summary and Contributions: A novel approach for making imitation learning work with deep learning was prevented that avoids the pitfalls of monolithic end-to-end approaches and instead leverages on a modular approach -- while remaining a deep learning approach.

Strengths: The paper is theoretically sound and - for NeurIPS as a machine learning conference - the evaluation in simulation is sufficient. The paper stands out in comparison to robotics work at NeurIPS.

Weaknesses: There are no real robot evaluations and more tasks could be tried.

Correctness: The paper appears correct.

Clarity: Easy and fun read.

Relation to Prior Work: The paper is well-embedded in the literature.

Reproducibility: Yes

Additional Feedback: I would recommend ganging up with a robotics group with sufficient hardware for real robot experiments -- but that's gotta be Post-COVID-19.


Review 4

Summary and Contributions: The paper proposes a hierarchical approach to bi-manual manipulation. The high-level model selects a primitive given previous states. The low-level controllers (primitives) predict following states given the current state. Low and high-level controllers are learnt with a supervised training from manually designed demonstrations of the task. The proposed approach allow to generalized to unseen initial states.

Strengths: The paper proposes an interesting the low-level generative model, which draws inspiration from the standard RNN encoder-decoder approach. The proposed solution leverages: (1) a graph attention layer (GAT) for representing strong correlation in the the arm joints; (2) a residual skip connection (RES) which emphasizes the role of the goal state (e.g. the position of the lifted object) over other state variables (e.g. the pose of the arm). Results (Table 1 and Table 2) provide ablation studies which suggest that all these components are necessary to improve the success rate on two different bi-manual tasks.

Weaknesses: Performance (i.e. success rate) on the more difficult task (peg-in-hole) are relatively low and videos seem to suggest that the definition of success for this task was quite generous. It seems therefore that the proposed approach doesn't generalize well to bi-manual tasks which require a lot of precision, or otherwise robustness with respect to the initial state of the environment.

Correctness: Yes

Clarity: Yes, the paper is well written. A few typo: Page 4, line 130 "dynamics function" -> "dynamic functions" Page 8, line 230 "star" -> "start"

Relation to Prior Work: Yes

Reproducibility: Yes

Additional Feedback: For some reason the submitted PDF is encrypted. As a result some features (e.g. copy-paste and search) are not possible on the provided PDF. This makes the review slightly less comfortable.

[Author Response · NeurIPS 2020]

We thank the reviewers for their insightful comments. We are glad reviewers acknowledge our promising approach of
using hierarchical control and relational information to generalize bimanual manipulation imitation learning **R1, R2,**
**R3, R4**. We are encouraged by comments about the soundness of our experiments and theory **R1, R3**. We will fix the
minor issues and typos in the updated version. Below we clarify our approach and address specific concerns.

**R1, R2** *Related work and baselines* To the best of our knowledge, using deep imitation learning for bimanual
manipulation have only involved MLP [1] and RNN [2]. [4] uses graph representation for visual imitation learning.
MLP did not suit our task as we wanted to predict trajectory sequences. The GRU-GRU model presented in Table 1
was an implementation of RNN, which we adapted to our simulation environment. Graph neural networks with skip
connections in the context of bimanual manipulation have not been previously investigated. We also experimented with
GAIL [3] but it did not perform well. Errors for the first primitive were at least 6 times higher than HDR-IL. This is
likely due to the low variances of our expert demonstrations, making the policy distribution difficult to fit.

**R1, R4** *29% accuracy rather low...* The success rate is low because the task is intended to test the limits. In our
high-precision table lifting task, we showed 100% success rate in Table 1. This demonstrates our HDR-IL model greatly
improved generalization compared to prior methods. To deepen our investigation, we designed the more challenging
peg-in-hole task to test multiple generalizations in far out (up to 130) time steps.

**R1** *...an error of 5cm with HDR-IL seems unreasonably high.* The Euclidean distance errors shown in Table 1 **are**
**normalized** to per datapoint position. These errors do not necessarily reflect the simulation task performance. Task
success is largely determined by the accuracy at the end of a primitive. Errors in our model tend to be at the beginning
as suggested by the trajectories in Figure 4. Furthermore, Table 2 of the Appendix show the 5cm error is largely driven
by the Extend primitive, which is less crucial for the task success. The Extend primitive has the largest step sizes
between datapoints, as illustrated in Figure 3(a). The bigger steps leads to bigger prediction errors.

**R1** *...actual pose of one arm...relative pose of the second arm...* We ran tests and found using relational data does not
generalize as well as graph structure with absolute poses. Euclidean distances were $8.15 \pm 4.71$ with success rate 16%
compared to $5.64 \pm 5.17$ and 72% for ResInt in Table 1. We will add these to our baseline in an updated version.

**R1** *...effect of orientation...* Orientation of the grippers were included in the prediction to improve inverse kinematics
accuracy. We will quantify orientation errors of the gripper quaternions in an updated version. The generalization of the
model to different table starting orientations is a topic for future study.

**R1** *Not clear if...specific to bimanual manipulation* Our method is NOT specific to bimanual manipulation, but the
complex dependencies in the bimanual manipulation setting highlight the value of graphs in our experiments.

**R2** *...why did the ground truth demonstration fail...* The failures in demonstration are due to approximations of the
inverse kinematics (IK) solver. Such failure cases are rare (<0.7% of 2500 demonstrations). The IK solver sometimes
outputs unusual movements between poses which cause the table to be dropped. The arm trajectory in the demonstration
will change accordingly. Our model can account for these uncertainties in demonstrations because of the stochastic
sampling design shown in Figure 2.

**R2** *...limited to the provision of the graph structure.* Providing the graph structure is a common way to encode inductive
bias e.g. [4], not a limiting factor. Learning the graph structure is related to model selection and causal reasoning, which
is out of scope of this paper. However, we did experiment with a few different graph structures before arriving at our
fully connected graph with attention.

**R2** *...does not show off how complex two handed settings...* Our simulations include complex **physical contacts** and
**friction**. We did not use a flexible table. The peg-in-hole task was specifically designed to increase the task complexity:
assembling the table before lifting requires more coordination between the two arms. We showed a failed attempt at
lifting the table using one arm in our supplementary videos. Doing the same for either piece in the peg-in-hole task will
have the same result. A hammer and nail task would be an even more complicated experiment for future work.

**R2** *...manually crafted primitives...contradicting to learning...* This is NOT a contradiction. The primitives are
constructed based on the control commands executed in simulation. During training, the primitives are labeled. Our
model learns to compose the primitives for high level planning. It also learns the low level dynamics for each primitive.
During inference, the model predicts the primitives based on the demonstration trajectories.

[1] R. Chitnis, et al., "Efficient Bimanual Manipulation Using Learned Task Schemas," *ICRA*, 2020

[2] D. Rakita, et al., "Shared Control–Based Bimanual Robot Manipulation," *Science Robotics*, 2019

[3] J. Ho, et al., "Generative Adversarial Imitation Learning," *NeurIPS*, 2016

[4] M. Sieb, et al., "Graph-structured Visual Imitation," *CoRL*, 2020


[Meta-Review · NeurIPS 2020]

The reviewers appreciated the difficulty of simulated bimanual robotics problem approached and the novelty of the solution presented. While multiple reviewers (especially R2) had some reservations about whether this approach would actually be useful on real robots, there was a consensus view that the approach could inspire further developments and the novel contributions outweighed the lack of real robot validation.